# Area deprivation, screen time and consumption of food and drink high in fat salt and sugar (HFSS) in young people: results from a cross-sectional study in the UK

Fiona Thomas,[1] Christopher Thomas,[1] Lucie Hooper,[1] Gillian Rosenberg,[1] Jyotsna Vohra,[1] Linda Bauld[2]

¹Cancer Policy Research Centre, Cancer Research UK, London, UK
²Usher Institute of Population Health Sciences and Informatics, University of Edinburgh, Edinburgh, UK

**Correspondence to**
Miss Fiona Thomas;
fiona.thomas@hotmail.com

## ABSTRACT

**Objectives** To investigate associations between deprivation in young people and consumption of foods high in fat, salt and sugar (HFSS), screen time exposure and health knowledge.

**Design** An online cross-sectional survey with people aged 11–19 years in the UK, where participants reported consumption behaviours across 13 HFSS and two non-HFSS groups; screen time for commercial television and streaming services; and knowledge of health conditions and their links to obesity.

**Setting** UK

**Participants** A total of 3348 young people aged 11–19 years across the UK.

**Main outcome measures** The study assessed the consumption behaviours, commercial screen time exposure and the health knowledge of 3348 people aged 11–19 years. Multivariate binary regression analysis, controlling for age and gender, was performed.

**Results** Deprivation level was associated with increases in consumption of six of the HFSS products including energy drinks (OR: 2.943, p<0.001) and sugary drinks (OR: 1.938, p<0.001) and a reduction in consumption in the two non-HFSS products included in the study, fruit (OR: 0.668, p=0.004) and vegetables (OR: 0.306, p<0.001). Deprivation was associated with high weekly screen time of both television (OR: 2.477, p<0.001) and streaming (OR: 1.679, p=0.001). Health knowledge was also associated with deprivation. There was lower awareness of the association of obesity and cancer (OR: 0.697, p=0.003), type 2 diabetes (OR: 0.64, p=0.004) and heart disease (OR: 0.519, p<0.001) in the most deprived.

**Conclusions** Young people from the more deprived areas of the UK were more likely to consume a range of HFSS products, report increased exposure to HFSS advertising and have a poorer awareness of health conditions associated with overweight and obesity. The findings suggest that population-level measures addressing childhood obesity should account for consumption patterns among different groups of children and young people and the factors that may influence these.

## Strengths and limitations of this study

► The study identified an association between socioeconomic deprivation and risk factors that may influence the prevalence of childhood obesity in the UK through a nationally representative sample of young people aged 11–19 years across England and each of the devolved nations.

► Data collected for this study were from a single cross-sectional survey, so it is not possible to determine causation between the variables.

► The study could not directly assess exposure to marketing of foods HFSS, although previous research that had explored the relationship between commercial screen time and exposure and found the two to be related provided us with greater confidence that screen time may be a reliable proxy for marketing exposure under current UK marketing regulations.

## BACKGROUND

In the UK, around 30% of children are overweight or obese,[1] the highest rate of childhood obesity in Europe. This overall figure masks considerable disparities by socioeconomic status. Overweight and obesity prevalence for children in the 10% most deprived areas in England, for example, are more than double that of those who live in the least deprived 10%.[2] Longer term, an obese child is around five times more likely to become an obese adult,[3] and there is substantial evidence that obesity in adulthood directly contributes to the development of conditions such as diabetes, coronary heart disease, stroke and 13 different types of cancer.[4–8]

Previous studies have provided diverse explanations for the rise in levels of obesity, ranging from genetics, increased calorie intake, an increase in sedentary behaviour or

a combination of factors.[9–11] The calorie intake increase is thought to be the most significant influence accounting for this rise, caused by a range of environmental factors,[9 12 13] including the role of the marketing and promotion of foods that are high in fat, salt and sugar (HFSS), often referred to as 'junk foods'. Marketing of these foods is extensive and delivered through a variety of platforms including television, streaming, price promotions and print media. Studies have identified a substantial expenditure by manufacturers and retailers on the marketing of junk food to children and young adults[14 15] and identified that industry recognises the potential marketing has for influencing consumption choices.[16] The link between marketing and weight outcomes, as well as increased consumption of HFSS products, has been highlighted by a number of previous studies.[17–23] Assessing exposure to HFSS marketing via self-reported recall of viewing advertisements does have limitations. Thus, some studies have used commercial screen time as a proxy for TV and online marketing exposure,[17 21] whereby greater screen time indicates increased exposure to HFSS advertising. Prior content analysis of UK television, where young people make up a large proportion of the audience, highlighted the increased likelihood of HFSS marketing exposure,[24] supporting this proxy measure.

Increasingly, there is a need to identify how marketing and promotion affect children in different social groups or those living in more or less deprived communities. Studies have previously identified an association between socioeconomic status and obesity.[25 26] This association is likely to be caused by a wide variety of factors including the pricing and availability of particular products in a locality,[27] understanding and awareness of dietary factors and weight and social norms.[28 29] However, further research is required to understand the inter-relationship of these factors and also other drivers of consumption including exposure to HFSS advertising. This type of research is important to identify interventions or policy actions that can contribute to addressing overweight and obesity. Halting or reversing current obesity trends is a current priority for public health policies in the UK,[30 31] and globally.[32 33]

Hooper et al[34] identified that there is a low level of public awareness of the link between overweight and obesity, resulting in preventable health conditions, including cancer. Only 25% of the UK adult population are aware of this link, and this lack of awareness is more prevalent in less affluent groups. Other studies have also found an association between greater health awareness and increased support for policy change, particularly for alcohol policy.[35 36] Greater health knowledge may therefore affect how young people view the acceptability of HFSS marketing and also consumption choices.

To date, there is limited research on the association between deprivation, HFSS marketing and obesity. Given these gaps, this study aims to investigate whether such a relationship exists and how it might be influenced by particular mediators such as frequency and duration of exposure to marketing and knowledge and understanding of health risks.

## METHODS

### Study design

An online cross-sectional survey was conducted between April and June 2017. The survey was developed following cognitive testing with a small sample of young people (n=100) to ensure age and cultural comprehensibility of the questions, some of which were based on well-validated questions used in other surveys.[21 37–40] This survey was compiled using the prevalidated questions and the advice of senior researchers from the National Cancer Institute, USA and Public Health England and the comprehensive guidance of the Institute for Social Marketing and the University of Stirling who have experience of running the Youth Tobacco Policy Survey. The final survey covered six main themes: exercise levels, food and drink consumption, screen time, recalled marketing exposure, perceptions of marketing and demographic factors.

A total of 3348 young people, aged 11–19 years, were recruited by market research company, YouGov, using their in-house panel. YouGov already had data on the children in households of adult in-house panel members. Children over the age of 16 years were directly approached and asked if they wished to participate. For those aged under 16 years, their parents were contacted and asked if their child could participate in the survey. Data collected were weighted by age, gender, ethnicity, region and social grade to be representative of the UK population.

### Measures

#### Deprivation

Level of deprivation was assessed using an area-based measure rather than individual measure of socio-economic status. The Index of Multiple Deprivation (IMD) was coded into five equal quintiles for analysis, ranging from 1, the lowest 20% of deprivation, to 5, the highest 20% of respondents. IMD is a measure of the relative deprivation of an area, combining information from seven domains: income deprivation, employment deprivation, education, skills and training deprivation, health deprivation and disability, crime, barriers to housing and services and living environment deprivation.[41]

#### Consumption behaviours

The survey measured consumption of a range of food and drink products. Participants were asked to report their consumption behaviours from the question 'How often do you usually eat or drink…?' followed by a series of food categories, HFSS including biscuits and cakes, chips, confectionary, crisps, desserts, energy drinks, flavoured yoghurts, milk-based drinks, ready meals, sugary drinks, sweetened cereal and takeaways, as well as with healthy items such as fruit and vegetables. These food groups were chosen using previous research on unhealthy products and with reference to the categories

included by Public Health England in their sugar reduction programme[42] and the nutrient profiling model to ascertain nutritional composition of foods. Responses were graded on a Likert scale from more than once a day to never and then converted to binary variables across two coding groups. The groups identified as 'higher' consumption depending on the total calorific content in each food.[17 21 42] The first coding group included sugar-sweetened drinks, flavoured yoghourts, confectionary, cakes/biscuits, fruit, vegetables, diet drinks, crisps and desserts where two or more portions a week was considered high consumption.[42] The second coding group included takeaways, ready meals, energy drinks, fried potato products, milk-based drinks and sugar-sweetened cereals where one or more portions a week was considered higher consumption.[42] This coding was deduced from a range of approaches including the Public Health England nutrient model, pilot testing and a review of the average calories in each product, recognising the differences between portion sizes of the food categories. The coding was only calculated for participants who gave an answer, and those who selected 'not sure' were excluded from the final analysis.

### Screen time

Commercial screen time was a variable created in the data set based on responses related to frequency and duration of exposure to TV and streaming (on demand) services.[21 43] Participants listed the hours spent watching both commercial and non-commercial TV and streaming services. This excluded screen time from computers being used for homework. Non-commercial screen time (which contains no paid for marketing in the UK context) was shown to not be significant in previous analysis of the data[17–19] and was therefore removed. Weekend and week-day viewing was then weighted and turned into a weekly measure and categorised; low (<3 hours per week), medium (3–21 hours per week) and high (21 hours or more per week).[21]

### Health knowledge

Health knowledge was assessed using the question 'Which, if any, of the following health conditions do you think can result from being overweight? Please tick all that apply.' Options included answers that were both correct and incorrect to identify the extent of health knowledge. The eight chosen conditions were cancer, stroke, heart disease, diabetes type 1, diabetes type 2, migraines, chicken pox and influenza. From this, the results were coded as a binary variable: 0—unaware; 1—aware of the links between certain conditions and being overweight.

### Age and gender

Control variables were selected on theoretical importance from a rapid review of the literature[21 40 43–46] and included gender (coded 0—male, 1—female) and age (11–19 years).

### Patient and public involvement

Prior to the development of the survey, qualitative research was carried out by colleagues at the University of Stirling and National Centre for Social Research. This work consulted young people through focus groups, on the design and content of the questionnaire. Results of these focus group discussions were published in a Cancer Research UK report 'It's Just There To Trick You'.[47] This Patient and Public Involvement (PPI) development work involved discussion of relevant research questions related to food and beverage consumption, relevant policy issues (ie, exposure to food marketing, pricing, availability), use of broadcast media examples and the appropriateness of questions relating to sociodemographic characteristics. Questions included in the resulting survey were then trialled with young people using cognitive interviewing techniques as described above. Survey reports are publicly available on Cancer Policy Research Centre's website.

## ANALYSIS

Data were analysed using IBM SPSS Version 23. Multiple multivariate binary regression models were run on the unweighted data to test for associations between deprivation levels and three key behaviours of young people—consumption behaviours, screen time use and health knowledge. The consumption model used the dependent binary variables of food and drink consumption behaviours. Models were run separately for each dependent variable, producing 15 models in total. The screen time model used the dependent variable of categorised reported screen time hours. The health knowledge model used the dependent binary variable of awareness of a health condition and its link to overweight and obesity. This included eight different health conditions, some with identified associations, and some without.

Within each of these models, the IMD variable was used as an independent variable, with the least deprived quintile as the reference group. Age and gender were included in the models as control variables, as potentially confounding variables.

## RESULTS

### Sample characteristics

Almost half (49%) of the survey respondents were female and 51% male. The mean age of participants was aged 14.9 years (SD=2.55). The majority (82%) were from white British backgrounds with 18% from other ethnic groups. The majority of respondents lived in England (82%); 5% of respondents lived in Wales, 8% in Scotland and 3% in Northern Ireland (table 1).

### Screen time

Of respondents, there were 31.9% in low, 57.1% in medium and 11.0% in high screen time categories for television viewing. For streaming services, there was

**Table 1** Sample demographics of the UK representative respondents

| Male | |
|---|---|
| 11—12 | 11.0% |
| 13—15 | 16.0% |
| 16—17 | 12.0% |
| 18—19 | 12.0% |
| Female | |
| 11—12 | 10.0% |
| 13—15 | 16.0% |
| 16—17 | 11.0% |
| 18—19 | 12.0% |
| Ethnicity | |
| White | 82.0% |
| BME | 18.0% |
| IMD | |
| 1,2 | 20.0% |
| 3,4 | 20.0% |
| 5,6 | 20.0% |
| 7,8 | 20.0% |
| 9,10 | 20.0% |
| Region | |
| North East | 4.0% |
| North West | 11.1% |
| Yorkshire & Humber | 8.5% |
| East Midlands | 7.3% |
| West Midlands | 9.3% |
| East | 9.3% |
| London | 12.7% |
| South East | 14.0% |
| South West | 8.2% |
| Wales | 4.7% |
| Scotland | 7.8% |
| Northern Ireland | 3.1% |

BME, Black and Minority Ethnicity; IMD, Index of Multiple Deprivation.

30.3% in low, 50.7% in medium and 19.0% in high screen time categories.

### Deprivation and consumption behaviours

The results of the binary logistic regressions showed an association between deprivation and higher consumption behaviours, for a range of HFSS food products (table 2). The most deprived young people were significantly more likely to consume energy drinks (OR=2.943, p<0.001), followed by sugary drinks (OR=1.938, p<0.001).

In contrast, analysis identified consumption of fruit and vegetables was inversely associated with more deprived groups. Fruit (OR=0.668, p=0.004) and vegetables (OR=0.306, p<0.001) were more likely to be consumed in lower frequency by the most deprived respondents, when compared with the most affluent respondents, as per the use of IMD. Therefore, these young people had a reduced likelihood of consuming the healthier options in higher quantities.

### Deprivation and screen time

Regression analysis found an association between deprivation in young people and high weekly screen time of both television and streaming (table 3). The model compared 'high' category screen time (21 hours or more a week) to 'medium and low' screen time (<21 hours a week) and found those from the most deprived quintile were significantly more likely to be in the high screen time category than the more affluent respondents, for both television (OR=2.477, p<0.001) and streaming (OR=1.679, p=0.001).

### Deprivation and health knowledge

The analysis identified an association between deprivation and poorer health knowledge (table 4). Respondents were asked whether eight health conditions (from a pre-existing list) could occur as a result of being overweight or obese. There was significantly poorer awareness of the association between cancer (OR=0.697, p=0.003), type 2 diabetes (OR=0.64, p=0.004) and heart disease (OR=0.519, p<0.001) and obesity for those from the more deprived quintiles. There was also significantly higher association between incorrectly linking type 1 diabetes (OR=1.536, p<0.001) and obesity in the most deprived quintile, compared with the most affluent quintile.

### DISCUSSION

Results from this survey identify a clear association between socioeconomic deprivation and risk factors that may influence the prevalence of childhood obesity in the UK. Involving a nationally representative sample of young people aged 11–19 years across England and each of the devolved nations, it sought to explore whether young people living in more deprived areas reported knowledge and behaviours that may contribute to obesity. The study found that these young people consumed more foods and beverages high in salt sugar and fat and were conversely less likely to report consumption of fruit and vegetables. In addition, young people living in more deprived communities spent more time watching commercial broadcast media where they could be exposed to HFSS advertising. Young people living in less affluent areas also had lower levels of awareness of the preventable health conditions, including cancer, which can arise as a result of obesity.

These results support findings from previous studies on factors influencing childhood obesity but also provide new evidence on the clustering of these factors among less affluent groups. It is well established that

**Table 2** Consumption behaviours and deprivation

| | Consumption behaviours | | | | | | |
| | Descriptive findings | | | | Logistic regression/significance | | |
| | Most deprived 20% | | Least deprived 20% | | Most deprived quintile | | |
| | High consumption (%) | Low consumption (%) | High consumption (%) | Low consumption (%) | OR | CI (95%) | value |
| Biscuits and cakes | 58.4 | 41.6 | 63.6 | 36.4 | 0.841 | 0.638 to 1.038 | 0.097 |
| Chips | 72.1 | 27.9 | 67.5 | 32.5 | 1.259 | 0.974 to 1.627 | 0.079 |
| Confectionary | 65.3 | 34.7 | 66.3 | 33.7 | 0.962 | 0.751 to 1.232 | 0.759 |
| Crisps | 62.8 | 37.2 | 58.5 | 41.5 | 1.232 | 0.965 to 1.572 | 0.093 |
| Desserts | 47.1 | 52.9 | 55.2 | 44.8 | **0.732** | **0.576 to 0.930** | **0.011** |
| Diet drinks | 35.7 | 64.3 | 31.2 | 68.8 | 1.233 | 0.959 to 1.585 | 0.102 |
| Energy drinks | 15.6 | 84.4 | 7.0 | 93.0 | **2.493** | **1.676 to 3.706** | **0.000** |
| Flavoured yoghurts | 28.0 | 72.0 | 27.3 | 72.7 | 1.061 | 0.812 to 1.386 | 0.067 |
| Fruit | 71.3 | 28.7 | 78.8 | 21.2 | **0.668** | **0.507 to 0.879** | **0.004** |
| Milk-based drinks | 31.3 | 68.7 | 22.6 | 77.4 | **1.613** | **1.229 to 2.118** | **0.001** |
| Ready meals | 64.3 | 35.7 | 56.3 | 43.7 | **1.416** | **1.111 to 1.712** | **0.005** |
| Sugary drinks | 41.6 | 58.4 | 27.2 | 72.8 | **1.938** | **1.506 to 2.494** | **0.000** |
| Sweetened cereals | 49.6 | 50.4 | 44.8 | 55.2 | 1.253 | 0.986 to 1.593 | 0.066 |
| Takeaways | 39.1 | 60.9 | 25.1 | 74.9 | **1.914** | **1.482 to 2.472** | **0.000** |
| Vegetables | 78.9 | 21.1 | 92.4 | 7.6 | **0.306** | **0.211 to 0.442** | **0.000** |

The bolded values indicate significance from the model.

there is a clear gradient in overweight and obesity by socioeconomic status in both adults and young people, with individuals from less affluent communities more likely to carry excess weight compared with their more affluent neighbours.[2 25 26 48–50] More limited research has explored how eating patterns vary by deprivation in young people. This study adds to existing evidence, suggesting that greater HFSS consumption and lower levels of fruit and vegetable consumption are more common in less affluent young people.[18 51]

This survey also asked young people about the time they spend watching television and on-demand screening services and calculated 'screen time' using an approach employed in previous studies.[17 21] While higher levels of screen time are associated with sedentary behaviour, which may contribute to obesity, they may also suggest greater exposure to broadcast media marketing including of HFSS foods. Previous research has found that children who spent more time watching commercial TV and on demand programmes in the UK are exposed to more HFSS food marketing than those with lower levels of screen time.[24 52] Viewing more HFSS advertisements on TV and streaming has been associated with higher HFSS consumption, with the difference between a high consumer and a low consumer being at least 520 junk food products a year.[17]

**Table 3** Screen time behaviour and deprivation

| | Screen time behaviours | | | | | | | |
| | Frequency | | | | | | Logistic regression/significance | |
| | Most deprived (20%) | | | Least deprived (20%) | | | Most deprived quintile | |
| | Low viewing (%) | Medium viewing (%) | High viewing (%) | Low viewing (%) | Medium viewing (%) | High viewing (%) | OR | CI | p value |
| Television screen time | 25.7 | 55.9 | 18.4 | 31.7 | 60.0 | 8.3 | **2.477** | **1.697–3.614** | **0.000** |
| Streaming screen time | 28.2 | 46.8 | 25.0 | 33.2 | 50.3 | 16.5 | **1.679** | **1.234–2.283** | **0.001** |

The bolded values indicate significance from the model.

**Table 4**  Health knowledge and deprivation

| | | Frequency | | Logistic regression/significance | |
|---|---|---|---|---|---|
| | **Overall awareness** | **Most deprived awareness %** | **Least deprived awareness %** | **OR** | **p value** |
| Cancer link | 42.0 | 36.1 | 44.4 | **0.697** | **[0.003]** |
| Heart disease link | 87.0 | 83.5 | 90.3 | **0.519** | **[0.000]** |
| Stroke link | 60.0 | 62.1 | 59.2 | 0.862 | [0.228] |
| Diabetes type 1 link | 39.0 | 46.1 | 36.3 | **1.536** | **[0.000]** |
| Diabetes type 2 link | 82.0 | 78.7 | 84.7 | **0.64** | **[0.004]** |
| Influenza link | 4.0 | 4.1 | 2.7 | 1.544 | [0.196] |
| Chicken pox link | 1.0 | 0.4 | 0.7 | 0.558 | [0.502] |
| Migraine link | 14.0 | 11.0 | 11.8 | 0.907 | [0.604] |

The bolded values indicate significance from the model.

We also found that young people living in more deprived areas had lower levels of awareness of the links between overweight and obesity and relevant health conditions. Awareness is relevant because evidence relating to other preventable risk factors (such as smoking and alcohol) suggest that health knowledge is relevant as a preliminary step toward changing behaviour, but also, importantly, understanding and support for policies and interventions that may address key factors that drive consumption including restrictions on marketing and pricing of unhealthy products.[35 36 53 54]

Taken together, our findings suggest that inequalities in rates of obesity in young people in the UK may be linked to knowledge and behaviours driven by key aspects of an obesogenic environment. Action to address childhood obesity needs to take into account differential consumption patterns among less affluent young people and the factors that may influence these consumption patterns. The introduction of policies and interventions that aim to address these factors, including better information on the health consequences of obesity, reducing exposure to HFSS marketing and other wider population level measures (such as policies to address the price and content of products) should consider and assess their impact on less affluent groups.

### Strengths and limitations of the study
This study has a number of limitations. Data are from a single cross-sectional survey, so it is not possible to determine causation. The measure of deprivation used was an area-based measure (IMD), which has limitations, as individual or household deprivation may vary within areas. Young people may have limited knowledge and awareness of different health conditions, for example, the distinction between type 1 and type 2 diabetes, and therefore, responses to the question relating to health conditions may in part reflect this lack of understanding. Responses to each of the key topics of interest including food consumption patterns, use of streaming services and TV viewing,

and awareness of health conditions linked to obesity were based on self-report and thus subject to misreporting or recall bias. The study could not directly assess exposure to HFSS marketing, although our previous research has explored the relationship between commercial screen time and exposure and found the two to be related,[17 21] which provides us with greater confidence that screen time may be a reliable proxy for marketing exposure under current UK marketing regulations. The response rate for this survey is estimated at 26% for people aged 16–19 years and 47% for people aged 11–15 years with the parental consent process. This is in line with what is usually obtained for surveys with young people by YouGov and is a limitation when sampling from this population demographic. Finally, overweight and obesity in young people are driven by a wide range of factors beyond those assessed in this study.

### Conclusions and future research
Taken together our findings suggest that inequalities in rates of obesity in young people in the UK may be linked to knowledge and behaviours driven by key aspects of an obesogenic environment. Action to address childhood obesity needs to take into account differential consumption patterns among less affluent young people and the factors that may influence these consumption patterns. The introduction of policies and interventions that aim to address these factors, including better information on the health consequences of obesity, reducing exposure to HFSS marketing and other wider population level measures (such as policies to address the price and content of products) should consider and assess their impact on less affluent groups.

Future research should explore in more detail a larger number of factors, including, for example, the affordability and availability of HFSS foods, social norms and the influence of social networks in more deprived communities and how these influence knowledge and behaviour among more deprived young people. In addition, studies should assess the impact of changes to the

policy and regulatory environment proposed in the UK and other counties to reduce childhood obesity and how these changes may affect young people living in communities where obesity rates are highest.

**Acknowledgements** The authors are grateful to the Cancer Policy Research Centre, Cancer Research UK for funding the study. The views expressed are those of the researchers and not necessarily those of the funder.

**Contributors** Fiona Thomas carried out the majority of the analysis for this publication along with Chris Thomas and Lucie Hooper who also developed the questions for the survey. Gillian Rosenberg contributed to the preparation of the manuscript and analysis plan. Both Jyotsna Vohra and Linda Bauld contributed to the study design and concept and the preparation of the manuscript.

**Funding** This study was funded by Cancer Research UK (http://dx.doi.org/10.13039/501100000289).

**Competing interests** None declared.

**Patient consent for publication** Not required.

**Ethics approval** Ethical approval was obtained in January 2017 from the General University Ethics Panel at the University of Stirling. This ethical approval covered both cognitive testing of the questionnaires and the online surveys. YouGov's staff included lead for ethical and quality assurance to ensure adherence to best practice throughout testing and data collection.

**Provenance and peer review** Not commissioned; externally peer reviewed.

**Data sharing statement** There is no additional unpublished data available.

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
