## [Reviewer comments · BMJ Open]

ARTICLE DETAILS

TITLE (PROVISIONAL)	Area deprivation, screen time and consumption of food and drink high in fat salt and sugar (HFSS) in young people: Results from a cross sectional study in the UK
AUTHORS	Thomas, Fiona; Thomas, Christopher; Hooper, Lucie; Rosenberg, Gillian; Vohra, Jyotsna; Bauld, Linda

VERSION 1 - REVIEW

REVIEWER	Alison Gustafson University of Kentucky
REVIEW RETURNED	13-Nov-2018

GENERAL COMMENTS	The authors have done an excellent job.
---

REVIEWER	Stephen Whybrow University of Aberdeen, Scotland.
REVIEW RETURNED	16-Jan-2019

GENERAL COMMENTS	This manuscript explores relationships among self-reported consumption of foods high in fat, salt and sugar, screen time exposure to the marketing of such products and selected health knowledge amongst 11-19 year olds in the UK. The reported associations are not unexpected and are consistent with the literature. The main limitation appears to be that body weight status is not reported in the manuscript (or by participants during data collection?). The manuscript relies on associations reported in the literature between the behaviours assessed in this study and obesity. It is possible (although admittedly unlikely) that participants in this study who were living in more deprived areas were not more overweight than those living in less deprived areas. Specific comments: Title: The title does not accurately describe the contents of the paper. It suggests an exploration among health inequalities, "junk food" marketing and prevalence of obesity, but measures of body weight status are not included, and health inequalities are assumed from the area based IMD. The title should be changed. Page 6 Line 20 "Junk food" is a pejorative and ill-defined term. If it is necessary to use it, it should be defined here, or at least linked to HFSS foods (e.g. page 6 line 18 "...foods that are high in fat, salt and sugar (HFSS), often referred to as "junk foods". " Page 7 Consumption behaviours. Why are diet drinks included in the HFSS category? Also, why do they contribute to the food
---

	group identified by higher consumption depending on total calorific content of each food? Diet drinks are not high in fat, sugar, salt or energy. Was any distinction made between "energy drinks" and "sugary drinks"? There appears to be an overlap of the two; many energy drinks will contain a lot of sugar. Page 7 Line 48 "two or more portions a week was considered high consumption." Page 7 Line 50 "one or more portions a week was considered higher consumption." How were these boundary values for high consumption derived? Perhaps from ref 39, if so please include the reference for this. Page 8 Lines 8 - 10 "Weekend and week-day viewing was then weighted and turned into a weekly measure..." How was this done? "and categorised..." The categories do not appear to be given the reference 21. Page 8 Line 17 "Flue" Page 8. Ethics. How was informed consent obtained? Please add. Page 12 Line 12 "... overweight and obesity in young people is driven by a wide range of factors beyond those assessed in this study." One of which is physical activity level. Exercise levels were reported (page 7 Line 10), yet this was not included in the analysis. Why not? A further limitation is that overweight and obesity was not assessed directly, only indirectly from other research showing a link between prevalence of obesity and levels of HFSS marketing and consumption. It is possible that participants in this study who were living in more deprived areas were not more overweight than those living in less deprived areas. Page 9 Line 24 I think it is incorrect to use an area-based measure of deprivation to make assumptions about individual households. Within a deprived small area there could be some households that relatively "well off" and the opposite in less deprived areas. Some of the text should be changed to avoid this assumption. For example, "... were more likely to be consumed in lower frequency by the most deprived respondents, when compared to the most affluent respondents" - you cannot assume that participants living in the most deprived areas were the most deprived, or that those living in the least deprived areas were affluent. Reference 14: "Alliance OH" should be "Obesity Health Alliance"
--	--

REVIEWER	S. Howard Wilsher UEA UK
REVIEW RETURNED	24-Jan-2019

GENERAL COMMENTS	Hi, this is a very interesting paper and well written. I have a few queries and comments. Line 17 - wrong flu spelling Line 25 - what area-based measure did you use for IMD? The size of this could make a large difference in results. Were the results
--

	checked for skewing given the response differences in regions. The Obesity Forum create a map of obesity levels in children - might be worth a look and compare with your results. See also Howard Wilsher - sales of healthy and unhealthy foods, which could be important for your study. Line 48 - two or more portions of fruit and vegetables per week is very low, not high. However, I understand your need to split the data somehow, but really this is a spurious segmentation, given that 5 a day is the recommended level. There was no mention of 5 a day in the introduction or discussion. Was a full regression model created to assess the impact of each variable? Might be interesting. Young children may not be aware of some of these health conditions, especially the difference between type 1 and 2 diabetes. They may have chosen 1 over 2 for perceived importance. A limitation as is online surveys. Ref 49 is an old reference - is there anything more up to date?
--	---

VERSION 1 – AUTHOR RESPONSE

Reviewer 1:

N/A

Reviewer 2:

Title: The title does not accurately describe the contents of the paper. It suggests an exploration among health inequalities, "junk food" marketing and prevalence of obesity, but measures of body weight status are not included, and health inequalities are assumed from the area based IMD. The title should be changed.

Our response: The title has now been changed to more accurately reflect the content of the paper

Page 6 Line 20 "Junk food" is a pejorative and ill-defined term. If it is necessary to use it, it should be defined here, or at least linked to HFSS foods (e.g. page 6 line 18 "...foods that are high in fat, salt and sugar (HFSS), often referred to as "junk foods").

Our response: A line has been added to link the term junk foods to the description of HFSS to ensure the reader understands the link.

Page 7 Consumption behaviours. Why are diet drinks included in the HFSS category? Also, why do they contribute to the food group identified by higher consumption depending on total calorific content of each food? Diet drinks are not high in fat, sugar, salt or energy.

Our response: Diet drinks were included in a previous paper (Thomas et al, 2018) but the reviewer is correct that they are not relevant to this analysis and have been removed.

Christopher Thomas, Lucie Hooper, Robert Petty, Fiona Thomas, Gillian Rosenberg and Jyotsna Vohra (2018) "10 Years On: New evidence on TV marketing and junk food eating amongst 11-19 year olds 10 years after broadcast regulations".

Was any distinction made between "energy drinks" and "sugary drinks"? There appears to be an overlap of the two; many energy drinks will contain a lot of sugar.

Our response: Building on a previous published analysis from CRUK (Thomas et al, 2018) the questionnaire identified separate examples from each food groups used and the calories for each, these are identified in the table on page 38, Table 2 of the paper.

Christopher Thomas, Lucie Hooper, Gillian Rosenberg, Fiona Thomas, Jyotsna Vohra. (2018) "Under Pressure: New evidence on young people's broadcast marketing exposure in the UK".

Page 7 Line 48 "two or more portions a week was considered high consumption."

Page 7 Line 50 "one or more portions a week was considered higher consumption."

How were these boundary values for high consumption derived? Perhaps from ref 39, if so please include the reference for this.

Our response: Thank you for pointing out this omission. The relevant reference has now been added to the two sentences, based on the categorisation from this prior referenced paper

Page 8 Lines 8 - 10 "Weekend and week-day viewing was then weighted and turned into a weekly measure..." How was this done?

Our response: Weighted to be representative of the week; as identified in prior CRUK paper (ref 17 & 18) and previously implemented in an Australian paper as referenced in the document. This Australian paper is reference 21 in the paper.

"and categorised..." The categories do not appear to be given the reference 21.

Our response: the idea to categorise the measure rather than scale was developed from reference 21. As reference 21 states 'A weighted average daily time spent watching commercial television was calculated and collapsed into the following categories: none; 2 h or less per day (moderate viewers); more than 2 h per day (high viewers).'

Page 8 Line 17 "Flue"

Our response: Amended

Page 8. Ethics. How was informed consent obtained? Please add.

Our response: Additional information has now been added to the section on ethics.

Page 12 Line 12. "... overweight and obesity in young people is driven by a wide range of factors beyond those assessed in this study." One of which is physical activity level. Exercise levels were reported (page 7 Line 10), yet this was not included in the analysis. Why not?

Our response: Physical activity was found to be insignificant in the univariate analysis initially conducted of the data. As a result it was excluded from the final regression analysis.

A further limitation is that overweight and obesity was not assessed directly, only indirectly from other research showing a link between prevalence of obesity and levels of HFSS marketing and consumption. It is possible that participants in this study who were living in more deprived areas were not more overweight than those living in less deprived areas.

Our response: Obesity has now been removed from the title to better reflect the analysis conducted.

Page 9 Line 24 I think it is incorrect to use an area-based measure of deprivation to make assumptions about individual households. Within a deprived small area there could be some households that relatively "well off" and the opposite in less deprived areas. Some of the text should be changed to avoid this assumption. For example, "... were more likely to be consumed in lower

frequency by the most deprived respondents, when compared to the most affluent respondents" - you cannot assume that participants living in the most deprived areas were the most deprived, or that those living in the least deprived areas were affluent.

Our response: We are familiar with the issue of individual vs area-based deprivation. However, this study involved young people and measures of individual deprivation commonly used with adults (educational attainment, income, housing tenure and other measures) are not appropriate for studies with youth. However, we have added text to the discussion section of the paper to make clear that an additional limitation was that deprivation was assessed using an area-based measure.

Reference 14: "Alliance OH" should be "Obesity Health Alliance" CRUK fix using endnote

Our response: Amended

Reviewer 3:

Line 25 - what area-based measure did you use for IMD? The size of this could make a large difference in results. Were the results checked for skewing given the response differences in regions. The Obesity Forum create a map of obesity levels in children - might be worth a look and compare with your results. See also Howard Wilsher - sales of healthy and unhealthy foods, which could be important for your study.

Our response: The IMD is an area based measure commonly used in the UK that employs several separate measures of deprivation to develop a composite score. Further information is available here: (https://assets.publishing.service.gov.uk/government/uploads/system/uploads/attachment_data/file/579151/English_Indices_of_Deprivation_2015_-_Frequently_Asked_Questions_Dec_2016.pdf)

We recognise the limitations of using an area-based measure of deprivation rather than individual measures, although these are difficult to employ for young people rather than adults as indicated in the response to the reviewer above. We have now added additional text to the limitations section of the discussion to highlight this issues.

Line 48 - two or more portions of fruit and vegetables per week is very low, not high. However, I understand your need to split the data somehow, but really this is a spurious segmentation, given that 5 a day is the recommended level. There was no mention of 5 a day in the introduction or discussion.

Our response: The categorisation was based on previous reports published by CRUK (reference numbers 17 and 18 in the document) that split high and low consumption based on the calorific content of the food. Unfortunately very few young people in the UK reach the 5 a day recommended level.

Was a full regression model created to assess the impact of each variable? Might be interesting

Our response: Univariate regressions were undertaken for each variables and then chosen to be used in the multivariate regressions based on their significance to the study

Young children may not be aware of some of these health conditions, especially the difference between type 1 and 2 diabetes. They may have chosen 1 over 2 for perceived importance. A limitation as is online surveys.

Our response: We thank the reviewer for this suggestion and relevant text has been added to the limitations of the paper.

Ref 49 is an old reference - is there anything more up to date?

Our response: We have now added a more up to date reference.

VERSION 2 – REVIEW

REVIEWER	Stephen Whybrow University of Aberdeen UK
REVIEW RETURNED	04-Mar-2019

GENERAL COMMENTS	It's not usual to use uncommon abbreviations in a title, and it might be better to replace HFHS with "junk food" or high-fat high-sugar foods.
--

REVIEWER	S. Howard Wilsher UEA, UK
REVIEW RETURNED	27-Mar-2019

GENERAL COMMENTS	Hi, it looks as though the authors, however, there are a few queries. Background Was any thought given to what children might purchase for themselves and what parents might purchase? There are numerous studies about the context - food shops, fast food outlets etc. A study about sales of healthy/unhealthy foods and associations with obesity is pertinent to this study: "The relationship between unhealthy food sales, socio-economic deprivation and childhood weight status: results of a cross-sectional study in England". There is also a lot of research regarding public awareness about the link between diet and health, which could be elaborated upon. Some people are more aware than others and it may not be linked to SES. Were the survey questions validated at all, apart from the pilot? There has been quite a lot of research in this area that could have been modified. Methods The data sample was weighted, so what was the reply rate? This could skew results. It is not clear how you linked the personal data of participants with the IMD? There are geographical information systems (GIS) systems that allow this. What size area did you use? There can be huge differences in IMD measures for large areas. Did you consider over or under reporting on the behavioural questionnaire? How did you identify "higher calorific" foods? How did you total caloric value for the foods? Why did you code the foods as you did? Three groups would have been better: All food in the first group, except fruit and vegetables - these should be separate, then the second group. Using only one or two portions per week seems rather arbitrary. Fruit and veg should be 5 a day. Participants probably ate the same breakfast - sugared cereal for 7 days - cut off of 1/2 portions seems odd. Results Were the responses representative of age, gender, SES, etc?
---

	Discussion Details in background might be helpful here. You cite SES here, but that was not considered in the data analysis or background. Cannot tell whether seeing adverts equates to eating the foods. Also, see background, other details could be used here, such as sales of foods, and context. Limitations There is a lot of evidence about health literacy - not just having health information but acting on it. Also, there is a lot about the level of reading skills at about the age of 10 years is way under the details given in health promotion, prescriptions etc. perhaps your questionnaire was written at a comprehension level too high for the young people?
--	---

VERSION 2 – AUTHOR RESPONSE

Reviewer(s)' Comments to Author:

Reviewer: 2

Reviewer Name: Stephen Whybrow

Institution and Country: University of Aberdeen, UK

It's not usual to use uncommon abbreviations in a title, and it might be better to replace HFHS with "junk food" or high-fat high-sugar foods.

Our response: The title has been changed to not include an abbreviation

Reviewer: 3

Reviewer Name: S. Howard Wilsher

Institution and Country: UEA, UK

Background

Was any thought given to what children might purchase for themselves and what parents might purchase? There are numerous studies about the context - food shops, fast food outlets etc. A study about sales of healthy/unhealthy foods and associations with obesity is pertinent to this study: "The relationship between unhealthy food sales, socio-economic deprivation and childhood weight status: results of a cross-sectional study in England".

Our response: We are familiar with this article by our colleagues at Cambridge and the University of East Anglia and agree it is useful for highlighting issues related to the local food environment and how it might contribute to overweight and obesity. We have added this reference to the introduction. However, unfortunately in our study we do not have data on what children purchase for themselves and what parent's purchase

There is also a lot of research regarding public awareness about the link between diet and health, which could be elaborated upon. Some people are more aware than others and it may not be linked to SES.

Our response: We thank the reviewer for this suggestion and have added two references to the introduction that relates to public awareness.

Were the survey questions validated at all, apart from the pilot? There has been quite a lot of research in this area that could have been modified.

Our response: For most of the survey validated questions were adapted and used wherever possible. Questions were adapted for our instrument from the following surveys: FLASHE37 (family life activity sun health and eating study), the last three iterations of the Youth Tobacco Policy Survey, the National Secondary Students' Diet and Activity survey (NASSDA) run in Australia, the National Diet and Nutrition Survey (NDNS) run by Public Health England (PHE) and the University of Stirling's survey on brand engagement amongst young people. Each of these was adapted in some way during cognitive interviews conducted by the Market Research Company, or otherwise to ensure policy relevance, age appropriateness and cultural validity. The most relevant sources for this report were FLASHE (consumption questions) and NASSDA (screen time questions).

Methods

The data sample was weighted, so what was the reply rate? This could skew results.

Our response: Looking through the data and speaking with the company that collected the data the response rate was estimated at 26% for 16-19s and 47% for 11-15s with the parental consent process. This is within the range that they would expect when collecting data of this types from this demographic of the population and we can insert a line about this in the limitations section of the paper

It is not clear how you linked the personal data of participants with the IMD? There are geographical information systems (GIS) systems that allow this.

Our response: One of the items of demographic information collected from participants by the company was postcode they had the software/system to convert this to IMD numbers. We received anonymised data without the postcodes and only the IMD numbers included for analysis.

What size area did you use? There can be huge differences in IMD measures for large areas.

Our response: The study was carried out across the UK and did not focus on one specific geographical location or size – we aimed to get representative numbers from each of the devolved nations and boosted samples in Scotland and wales so that they were proportionally representative.

Did you consider over or under reporting on the behavioural questionnaire?

Our response: This is something we have already included in the limitations section of the article – we recognise this is a limitation when a survey relies on self-report (page 9).

How did you identify "higher calorific" foods? How did you total caloric value for the foods?

Our response: Within the methodology under consumption behaviours' the calorific food groupings were identified in line with the previous report authored by our team at CRUK (ref 17 and 18) and with reference to the Public Health England sugar reduction programme paper (ref 42) in the UK.

Why did you code the foods as you did? Three groups would have been better: All food in the first group, except fruit and vegetables - these should be separate, then the second group. Using only one or two portions per week seems rather arbitrary. Fruit and veg should be 5 a day. Participants probably ate the same breakfast - sugared cereal for 7 days - cut off of 1/2 portions seems odd.

Our response: The coding for the food groups were based on the Public Health England food categories and the guidance they provided (ref 42) as from previous report authored by our team at CRUK (ref 17 and 18).

Results

Were the responses representative of age, gender, SES, etc?

Our response: Yes, the full demographic information is available in Table 1 in the document.

Discussion

Details in background might be helpful here. You cite SES here, but that was not considered in the data analysis or background.

Our response: We do refer to the relationship between SES and obesity in the background section and in the methods section make clear that we used an area-based measure to examine SES (further explanations of the limitations of area-based measures are also included).

Cannot tell whether seeing adverts equates to eating the foods.

Our response: The methods used in our study don't claim causality (i.e. viewing adverts directly cause HFSS food consumption) but instead an association, as we explain in both the results and the discussion.

Also, see background, other details could be used here, such as sales of foods, and context.

Our response: As we mention above, there are multiple factors that determine what foods young people eat and what causes HFSS food consumption. We have expanded the introduction section and added references, including those suggested by the review above, to acknowledge the multi-faceted causal factors. However our article did not focus on food sales or use sales data.

Limitations

There is a lot of evidence about health literacy - not just having health information but acting on it. Also, there is a lot about the level of reading skills at about the age of 10 years is way under the details given in health promotion, prescriptions etc. perhaps your questionnaire was written at a comprehension level too high for the young people?

Our response: The questions used in our survey were developed with young people and cognitively tested with a sample of 100, including younger children. As outlined in the methods, this was to ensure age and cultural comprehensibility of the questions. While we cannot assume that all younger children in the final sample of 3,348 had the same level of literacy or indeed health literacy, the development phase of the survey did aim to test the question for precisely this reason, and indeed the cognitive testing resulted in a number of amendments and simplification of terms and wording. We followed best practice by undertaking cognitive testing before administering the survey because we were aware that the questions needed to be easily comprehensible to young people of different ages.